# Development of Miniaturized Water Quality Monitoring System Using Wireless Communication

**DOI:** 10.3390/s19173758

**Published:** 2019-08-30

**Authors:** Hsing-Cheng Yu, Ming-Yang Tsai, Yuan-Chih Tsai, Jhih-Jyun You, Chun-Lin Cheng, Jung-How Wang, Szu-Ju Li

**Affiliations:** 1Department of Systems Engineering and Naval Architecture, National Taiwan Ocean University, 2 Pei-Ning Road, Keelung 20224, Taiwan; 2Material and Chemical Research Laboratories, Industrial Technology Research Institute, 195, Sec. 4, Chung Hsing Road, Chutung, Hsinchu 31040, Taiwan

**Keywords:** embedded system, water quality monitoring, wireless communication

## Abstract

Recently, environmental pollution resulting from industrial waste has been emerging in an endless stream. The industrial waste contains chemical materials, heavy metal ions, and other toxic materials. Once the industrial waste is discharged without standards, it might lead to water or environmental pollution. Hence, it has become more important to provide evidence-based water quality monitoring. The use of a multifunctional miniaturized water quality monitoring system (WQMS), that contains continuous monitoring, water quality monitoring, and wireless communication applications, simultaneously, is infrequent. Thus, electrodes integrated with polydimethylsiloxane flow channels were presented in this study to be a compound sensor, and the sensor can be adopted concurrently to measure temperature, pH, electrical conductivity, and copper ion concentration, whose sensitivities are determined as 0.0193 °C/mV, −0.0642 pH/mV, 1.1008 mS/V·cm (from 0 mS/cm to 2 mS/cm) and 1.1975 mS/V·cm (from 2 mS/cm to 5.07 mS/cm), and 0.0111 ppm/mV, respectively. A LoRa shield connected into the system could provide support as a node of long range wide area network (LoRaWAN) for wireless communication application. As mentioned above, the sensors, LoRa, and circuit have been integrated in this study to a continuous monitoring system, WQMS. The advantages of the multifunctional miniaturized WQMS are low cost, small size, easy maintenance, continuous sampling and long-term monitoring for many days. Every tested period is 180 min, and the measured rate is 5 times per 20 min. The feedback signals of the miniaturized WQMS and measured values of the instrument were obtained to compare the difference. In the measured results at three different place-to-place locations the errors of electrical conductivity are 0.051 mS/cm, 0.106 mS/cm, and 0.092 mS/cm, respectively. The errors of pH are 0.68, 0.87, and 0.56, respectively. The errors of temperature are 0.311 °C, 0.252 °C, and 0.304 °C, respectively. The errors of copper ion concentration are 0.051 ppm, 0.058 ppm, 0.050 ppm, respectively.

## 1. Introduction

Industrial waste contains toxic substances such as chemical substances, which cause water and environmental pollution. Once the industrial waste is discharged without standards, it might lead to water or environmental pollution. Thus, it has become more important to inquire into water quality monitoring. With the progress of semiconductor, large data processing, and wireless communication technologies, a miniaturized water quality monitoring system (WQMS) could be developed in this study. Generally, in order to identify the level of water pollution, water quality indicators are used to analysis the water properties: physical (e.g., temperature, odor, turbidity, and electrical conductivity), chemical (e.g., pH value, copper ion concentration), and biological (e.g., content of organisms) characteristics. Among them, the water temperature indirectly affects some measured standards of the indicators, such as electrical conductivity, dissolved oxygen, and so on. Acidity is also called the pH value, which is the hydrogen ion concentration in water. Aquatic organisms are sensitive to acid and alkali, so the pH value is crucial to the survival of aquatic organisms. Electrical conductivity, related to the ionic strength of water, can be used to infer the quality of dissolved materials in water. It can be used to determine whether water is polluted or to what extent saltwater infusion occurs in a coastal area. The copper ion concentration in general water is between 20 and 75 ppb [1], but there may be a large amount of copper ions of water in old pipelines or contaminated water sources. If such water becomes household drinking water, it may affect human health. Hence, the US Environmental Protection Agency has made a standard policy to limit the copper ion concentration in lakes and rivers to less than 1 ppm, and drinking water must be less than 1.3 ppm [2]. Zhou et al. published a proposed low cost and miniature sensor chip with multi-parameters [3]. The sensor chip can detect the conductivity, temperature, and pH of water simultaneously. Iridium oxide film was electrodeposited as pH-sensing material to fabricate a chip. The linearity of the pH sensor was R^2^ = 0.9997 and sensitivity was −67.60 mV/pH. Furthermore, LoRa technology features a long range, low power consumption (i.e., long battery life), multi-node network, and low cost [4]. Otherwise, ZigBee technology is a low-power LAN protocol based on the IEEE802.15.4 standard. ZigBee is a two-way wireless communication technology with a closed range, low complexity, low power consumption, low speed, and low cost. It is mainly used for data transmission between various electronic devices over a short distance with low power consumption and a low transmission rate, and has typical applications with periodic data, intermittent data and low response time data transmission. It can be seen that the transmission distance can be lengthened by using LoRa technology, so LoRa has been chosen to adopt in this study. According to the National Environmental Water Quality Monitoring Information Network of Environmental Protection Administration, Executive Yuan, R.O.C. (Taiwan), it is necessary to directly measure the pollutant content in the water environment, including physical, chemical and biological data, to provide accurate and reliable data to describe the water quality, and to compare with the water quality standards of the water body for reference to the assessment of the conformity of water quality standards. According to the regulation of the National Environmental Water Quality Monitoring Information Network of Environmental Protection Administration, Executive Yuan, R.O.C. (Taiwan), most industrial areas have to meet the specified values. For example, the temperature must be below 38 °C, the pH measurement range should be from 6.0 to 9.0, and the copper ion content should be 3.0 ppm or less [5,6,7]. Based on the above reasons, this research proposes a miniaturized WQMS with multiple functions for wireless communication application of water quality monitoring.

## 2. Measurement Principle

The multifunctional WQMS using wireless communication technology is presented in this study, and it can be utilized to measure temperature, pH, electrical conductivity, and copper ion concentration, simultaneously.

### 2.1. Measurement of Temperature

The initial capacitance of the capacitive sensor is very small. The stray capacitance of the measuring circuit and the capacitance of the sensor plate and its surrounding conductors are large, which reduces the sensitivity of the sensor; on the other hand, these capacitors often change randomly, which can destabilize the sensor and affect measurement accuracy [8].

A resistance temperature detector (RTD) has many advantages of natural properties of the metal, including high stability, reliability and accuracy. When the temperature rises due to heat, the resistance of the metal will rise. Most RTDs are made of materials such as white gold, copper or nickel. The thin film type RTD is coated with a metal film on a substrate. The characteristics of RTD sensing resistance with temperature variation can be applied to measurement of temperature. The functional relationship between resistance value and temperature is in form of:(1)$$R_{T}=R_{0}\left(1+\alpha T\right)$$
where *R_T_* is the resistance value of RTD at *T* °C, *R*_0_ is the resistance value of the RTD at 0 °C, *α* is the temperature coefficient of the resistance, and *T* is the temperature in Celsius.

### 2.2. Measurement of pH Value in Water

The architecture of ion sensitive field effect transistor (ISFET) was set on removing the gate of the metal-oxide-semiconductor field-effect transistor (MOSFET) [9]. In the transistor, the currents are varied by the concentration of hydrogen ions in the solution when elements are soaked into the solution. The most serious shortcoming of the ISFET architecture is that the ion selection effect is poor, and the small size is difficult to process. When the measurement is performed, the immersion of the transistor in the solution is easily eroded by the object to be tested, resulting in damage or electrical changes [10]. Aiming to deal with the shortcoming of ISFETs, in 1983, Jan van der Spiegel et al. brought forth the architecture of the extend-gate field-effect transistor (EGFET) [11]. The main change was to keep the metal gate of the MOSFET separate from the gate and extend out with the wire, which required a low-impedance and a conductive film with high conductivity. The most importance part of the adjustment was to separate the sensing block from the component block, which improved the problem of vulnerability and damage. Moreover, the sensing area became adjustable in dimension and shape, which greatly improved the adaptability of the measurement [12].

### 2.3. Measurement of Electrical Conductivity

Electrical conductivity is a physical quantity that refers to the ability of an object to pass current. The rate of movement of ions is affected by temperature. The temperature is higher, and the movement of ions is faster. The solution carries a current through the movement of anion and cation. Thus, the electrical conductivity decreases with a temperature increase. Generally, differences in composition of water cause different temperature effects, and each increase of 1 °C causes the electrical conductivity to increase by 1.9%.

The two-pole electrode contains two parallel plate electrodes, and it applies alternating current at both ends. The voltage is measured between the two electrodes, so that the resistance of the solution can be determined. The reciprocal of the resistance is the electrical conductivity. Additionally, the conductance influence can be separated by using AC method, and it can measure the exact solution resistance [13].

### 2.4. Measurement of Copper Ion Concentration

Copper is an important trace element needed to sustain human life, but the amount should not be too great. When the copper ion content in water reaches 0.01 ppm, the self-purification ability of the water itself is suppressed. When the copper ion content exceeds 3 ppm, the water has a significant odor. Besides, the water is unsuitable to drink if the copper ion content exceeds 15 ppm.

Ion selective electrodes, also known as thin film electrodes, are an electrochemical type of sensor. The ion selective electrode can measure the potential of a specific ion. When a specific ion in the solution to be tested is in contact with the sensing film, a film potential is generated on the sensing film. This film potential can generate a potential difference from the potential of the reference electrode, which can be adopted to convert the concentration of the ion to be measured one.

### 2.5. Measurement of LoRa Wireless Transmission

LoRa technology has been developed as a long-distance wireless communication technology in recent years. Its transmission range is about 15 km, and the battery power supply of the wireless communication device can be used for several years. Additionally, low power wide area network (LPWAN), a power-saving and long-distance transmission method, was created in response to the demand for the wireless communication. Furthermore, long range wide area network (LoRaWAN) is one of a variety of LPWAN specifications that focuses on low-power and long-range transmission, which defines the communication architecture of the system architecture and network.

## 3. Experimental Setup

The miniaturized WQMS consists of four modules, which are the water quality monitoring module, control circuit module, wireless transmitter module, and the sample and cleaning module, as shown in Figure 1. The front and back views of the experimental device diagram of the real WQMS are shown in Figure 2.

### 3.1. Water Quality Monitoring Module

The designed sensing electrodes of the water quality monitoring module integrate three measured parameters including temperature, electrical conductivity, and pH values. The temperature indication used a gold-plated electrode to make an RTD-type temperature sensor, and its characteristics have good linearity. The conductivity indication was made of a two-pole electrode, and the platinum electrode was used in the measurement area, which can reduce the possibility of reacting with the object tested. The material quality of the platinum electrodes have a degree of protection for the sensing electrodes of the electrical conductivity. The pH sensing electrode was made of indium tin oxide (ITO) as the sensing film material, and the reference electrode was made of the non-reactive platinum electrode. Many studies of sensing terminals are based on commercially available stable pen-type sensors. This study mainly achieved the same effect as commercially available sensors at a low cost. In general, the reference electrode is configured with a glass film. Herein, the reference electrode was configured with an ITO film [14]. The gold-plated electrodes at the contact point could produce good conductivity and reduce noise. A part of the flow channel adopted a positive engraved acrylic mold, and it only needed to be filled with a proper amount of polydimethylsiloxane (PDMS). After drying, it could be combined with the sensing electrode. The copper ion electrode adopted a commercially available copper ion electrode. The electrode was an ion selective electrode in the form of a solid membrane, and the measurement concentration range was 5 × 10^−8^ M to 5 × 10^−1^ M. All sensing electrodes were made on a glass substrate and three layers of material were formed, whereof the first layer was ITO, the second layer was plated with platinum in a place other than the pH sensing region, and the third layer was a layer of gold on the contacts and the temperature sensor. The water quality detecting chip was cemented as shown on the left-hand side in Figure 3, and the copper ion selective electrode for monitoring copper ion concentration is shown on the right-hand side in Figure 3.

### 3.2. Control Circuit Module

The temperature monitoring was a gold-plated electrode in the form of RTD, which changed its resistance value with temperature. The RTD resistor was combined with the operational amplifier and used a 5 V operational voltage for the inverting input. One of the inverting amplifiers was converted to the RTD resistor. The output voltage could be changed with temperature, but the ATmega32U4 could not be measured. The output voltage value of an analog-to-digital converter (ADC) from the temperature measurement sub-circuit read by a microcontroller unit (MCU), *V_TEMP_*, can be expressed as
(2)$$V_{TEMP}=V_{CC}\cdot \frac{-R_{T}}{R_{T1}}\cdot \frac{-R_{T3}}{R_{T2}}$$
where the *R_T_* is the resistance value of the RTD, and *R_T_*_1_ is the resistance value of the first-stage operational amplifier circuit in the temperature measurement sub-circuit. *R_T_*_2_ and *R_T_*_3_ are the input resistance and feedback resistance of the second-stage operational amplifier circuit in the temperature measurement sub-circuit, respectively.

The pH measurement circuit adopts the architecture of EGFET for circuit design, and the IC was CD4007. The IC contains three sets of CMOS transistors. In this study, the gate, drain and source of the three sets of transistors were connected in parallel, which can change the characteristic curve, so that the output current increases, and the sensitivity increased when the pH value was measured. The pH measurement output voltage *V_pH_* was in form of:(3)$$V_{pH}=I_{D}\cdot R_{pH1}\cdot \frac{-R_{pH3}}{R_{pH2}}$$
where *I_D_* is the drain current, *R_pH_*_1_ is the resistance value of the current-to-voltage converter, *R_pH_*_2_ and *R_pH_*_3_ are the resistance values of the input resistance and feedback resistor of the operational amplifier circuit in the pH measurement sub-circuit, respectively. The pH measurement sub-circuit was designed with a second-order low-pass filter before the output. The purpose was to filter out the noise that may be caused by the 1 kHz AC signal used in the electrical conductivity measurement. The filtering power of the second-order low-pass filter was closer to the ideal filter than the first-order low-pass filter, which could provide better filtering effect.

When the conductivity sensor was filled with solution, the resistance between the two electrodes would decrease as the conductivity increases, and it was used as the resistance of the inverting input of the inverting amplifier [15]. The distance between the two electrodes in the solution was fixed, and the equivalent resistance and electrical conductivity of the solution were reciprocal. The output voltage of the secondary inverting amplifier *V_out_* was in form of:(4)$$V_{out}=V_{in\_EC}\cdot \frac{-R_{EC3}}{R_{EC}}$$
where *R_EC_* is the equivalent resistance between the A and B terminals of the conducting electrode, and *R_EC_*_3_ is the feedback resistance of the second stage of the conducting measuring circuit. The *V_out_* is an AC voltage signal, which must be converted to a DC signal to supply the analog input pin of the MCU. In this study, a single-phase half-wave rectifier was used to convert AC to DC and remove noise through a low-pass filter. The DC voltage *V_dc_* output from the single-phase half-wave rectifier was the input AC voltage, and the average value was in form of:(5)$$V_{dc}=\frac{1}{2\pi }\int_{0}^{\pi }\sin\left(\omega t\right)\;d\left(\omega t\right)=\frac{V_{\max}}{2\pi }\left[-\cos\left(\omega t\right)|{}_{0}^{\pi }\right]=\frac{V_{\max}}{\pi }$$
where *V*_max_ is the peak voltage of the AC voltage, *V_dc_* is the average voltage of the DC voltage output by the single-phase half-wave rectifier, and *V_dc_* is obtained by the first-order low-pass filter to obtain the ADC voltage value (*V_EC_*) of the conductivity measurement sub-circuit read by the MCU.

The copper ion measurement used the ion selective electrode, and the signal was transmitted to the measurement circuit by the BNC connector. The theoretical formula for calculating the output voltage of the copper ion measuring circuit, *V_Cu_*, is in the form of:(6)$$V_{Cu}=V_{in\_Cu}\cdot \left(1+\frac{R_{Cu1}+R_{Cu2}-R_{v}}{R_{Cu3}+R_{v}}\right)\cdot \left(1+\frac{R_{Cu5}}{R_{Cu4}}\right)$$
where *V_in_Cu_* is the ion-selective electrode measuring voltage signal, *R_Cu_*_1_ and *R_Cu_*_3_ are the resistances of the first-stage amplifier of the copper ion measuring sub-circuit, respectively, *R_Cu_*_4_ and *R_Cu_*_5_ are the resistances of the second-stage amplifier of the copper ion measuring sub-circuit, respectively, and *R_Cu_*_2_ is the total resistance of the variable resistor, and *R_v_* is the resistance of the variable resistance between sliding contact and the fixed contact of the *R_Cu_*_3_ terminal.

The wireless communication test of the LoRa module utilized the LoRa module from Dragino to communicate with the LoRa gateway LG-01. Due to the lack of a suitable cloud platform, this paper did not test the cloud platform upload program for the time being. The LG-01 was set to the WiFi receiver network mode. The network sent the wireless network signal through the router as the AP, and the LG01 acted as the receiver of the wireless network and connected through the router. Since the LG01 could not be used as both the receiving end and the AP, it could only be connected to the personal computer via the local area network, and the personal computer was connected to the Internet using the Ethernet route. The LG-01 and the personal computer were also connected to the same group of WiFi area networks. The LoRa module was connected to the personal computer through the UNO module below to compose and burn the firmware. The LoRa gateway was the server end. After receiving the request of the LoRa communication module, it could reply the command to the communication module indicating that it had been received, and the LoRa communication module sent the request to the LoRa gateway. If there was a gateway receiving the signal, it could receive a response from the LoRa gateway. If there was no error, the wireless communication between the two used the A-type transmission mechanism, and the message could be received at any time, which has the most power-saving advantages. Although the transmission waiting time was relatively long, the system did not require too much transmission efficiency. When it is placed outdoors for a long time, it can save power, which can increase the time that the whole system can operate, and reduce the repair numbers and cost of maintenance. Only the wireless communication between the module and the gateway was tested in this study. If the appropriate cloud platform or application program could be obtained during the application, the data could be uploaded and monitored as the wireless communication node.

## 4. Experimental Results

Three various local water sources were sampled for testing in this study, called Samples A, B, and C, respectively. The sampled water characteristics were tested by the instrument, and then monitored by the miniaturized WQMS. The continuous monitoring time was 180 min, and the measuring rate was five times per 20 min.

### 4.1. Monitoring of Temperature

The verification of temperature electrode of the miniaturized WQMS depended on the values between the AVR voltage of the MCU on the measured motherboard and real temperature. Temperature measured ranged from 15 °C to 50 °C, and was modified by used of a hot plate, while monitoring the temperature by a thermocouple simultaneously. The experimental setup for measuring temperature is shown in Figure 4.

The temperature electrode of the miniaturized WQMS showed a sensitivity of 0.0193 °C/mV in the range of 15 °C to 50 °C, and a good linearity (coefficient of determination R_T_^2^ = 0.9995) was found. The accuracy of temperature was ±0.193 °C. During heating, the temperature electrode was put into the water, and it recorded the AVR voltages of the MCU on the measured motherboard, and monitored the heating temperature of the water, simultaneously. As the results of temperature tests, an X–Y scatter diagram showing the calibration curve is shown in Figure 5.

The experimental results of temperature by using Samples A, B, and C are shown in Figure 6. The errors of temperature were 0.311 °C, 0.252 °C, and 0.304 °C, respectively. The temperature measured by the water quality testing system in both Sample A and Sample B was consistent with the temperature at which the laboratory had air conditioning. Sample C was tested in a non-air-conditioned state, and the error was not greater than 0.5 °C.

### 4.2. Monitoring of pH Value

To test the potential of the ITO film, an Arduino Leonardo was utilized in this study to record ADC values and save the monitoring data into the personal computer. The experimental setup for measuring pH values is shown in Figure 7.

The verification of pH calibration compared the AVR voltages of the MCU on the measured motherboard and real pH values. According to the standard solution pH 4, pH 7, and pH 10, respectively, the measured ranges were from pH 4.0 to pH 10.0. The accuracy of pH value was ±1.0917. The pH electrode of the miniaturized WQMS showed a sensitivity of −0.0642 pH/mV and had a good linearity (coefficient of determination R_pH_^2^ = 0.9954). The pH tests used the Arduino Leonardo to acquire the AVR voltages of the MCU on the measured motherboard, which were the electric potential of the ITO film; the calibration curve of pH values is shown in Figure 8.

The experimental results of pH values for Samples A, B, and C are shown in Figure 9. The errors of pH were 0.68, 0.87, and 0.56, respectively. The water quality detection system measured mostly weakly acidic values. Only a small number of data points were weakly alkaline. The measurement results were stable.

### 4.3. Monitoring of Electrical Conductivity

The experimental setup for measuring electrical conductivity is shown in Figure 10. The titration method was adopted to verify the conductivity calibration tests. Using the method, deionized water (DI water) was added into saturated salt water (electrical conductivity = 5.07 mS/cm) until the voltage was close to zero (electrical conductivity = 0.005 mS/cm). During the titration, using the portable conductivity device for testing, the values of the electrical conductivity and the AVR voltage of the MCU on the measured motherboard were recorded simultaneously. The experimental setup for measuring electrical conductivity is shown in Figure 10.

Taking the DI water as a sample, the saturated salt water was utilized by the titration method until the voltage reached the limit of 4 V. When the voltage reached 4 V, the DI water was adopted by the titration method again until the voltage returned to the limit of 0 V. During the titration process, the voltages and electrical conductivity values were recorded simultaneously. The accuracy of the electrical conductivity was ±0.0087 mS/cm from 0 mS/cm to 2 mS/cm. The accuracy of the electrical conductivity was ±0.0327 mS/cm from 2 mS/cm to 5.07 mS/cm. As the results of tests, the electric conductivity had a discontinuity at 2 mS/cm. We obtained two approximations for the electric conductivity. One had a sensitivity of 1.1008 mS/V·cm from 0 mS/cm to 2 mS/cm with the determination coefficient of its linearity R_EC1_^2^ = 0.9926; and the other had a sensitivity of 1.1975 mS/V·cm from 2 mS/cm to 5.07 mS/cm with the determination coefficient of its linearity R_EC2_^2^ = 0.5792, as shown in Figure 11.

The experimental results of electrical conductivity by using Samples A, B, and C are shown in Figure 12. The errors of electrical conductivity were 0.051 mS/cm, 0.106 mS/cm, and 0.092 mS/cm, respectively. The pollution of Sample A was lower than Sample B and Sample C. The lower value meant an upstream source and the higher value meant downstream. The values could get indicate higher pollution from factories for downstream sources and lower pollution from upstream sources.

### 4.4. Monitoring of Copper Ion Concentration

The copper ion selective electrode was adopted for the measuring experiment of copper ion concentration, as shown in Figure 13. The calibration curve of copper ion concentration is shown in Figure 14. The verification of the calibration curve of copper ion concentrations were respectively 10.0 ppm, 7.5 ppm, 5.0 ppm, 2.5 ppm, and 1.0 ppm, so the measured ranges were from 1 ppm to 10 ppm. The copper ion selective electrode for measuring copper ion concentration showed a sensitivity of 0.0111 ppm/mV and the determination coefficient of its linearity R_Cu_^2^ = 0.9255. The accuracy of copper ion detection was ±0.1846 ppm. During measuring, the signal of the copper ion selective electrode was transmitted to the copper ion measuring circuit; furthermore, the monitoring data from the copper ion measuring circuit via the measuring motherboard were recorded into the personal computer simultaneously.

The experimental results of copper ion concentration by using Samples A, B, and C are shown in Figure 15. The errors of copper ion concentration were 0.051 ppm, 0.058 ppm, 0.050 ppm, respectively.

It is speculated that there may have been no copper ion component or that the copper ion concentration was too low, possibly lower than the detection limit of the instrument. Disturbances may occur during the measurement process, affecting the readings of the water quality detection system, but most of the measurement values were zero. The miniaturized WQMS was designed to replace the portable instrument. Even though two peaks of copper ion concentration up to 3.0 ppm were detected by of the miniaturized WQMS, they might have been just some shot noise or disturbances. They were all under the regulation standard of the National Environmental Water Quality Monitoring Information Network of Environmental Protection Administration, Executive Yuan, R.O.C. (Taiwan).

## 5. Conclusions

A multifunctional miniaturized water quality monitoring system (WQMS) has been developed in this study to simultaneously measure temperature, pH, electrical conductivity, and copper ion concentration, and their sensitivities are 0.0193 °C/mV, −0.0642 pH/mV, 1.1008 mS/V·cm (from 0 mS/cm to 2 mS/cm) and 1.1975 mS/V·cm (from 2 mS/cm to 5.07 mS/cm), and 0.0111 ppm/mV, respectively. Furthermore, three different place-to-place comparisons had been sampled to be water quality samples (i.e., Samples A, B, C) in this study. The continuous sampling period is 180 min, and the measured rate is five times per 20 min. The feedback signals of the WQMS and measured values of the instrument were obtained to compare the difference. In the measured results at three different locations, the errors of electrical conductivity are 0.051 mS/cm, 0.106 mS/cm, and 0.092 mS/cm, respectively. The errors of pH are 0.68, 0.87, and 0.56, respectively. The errors of temperature are 0.311 °C, 0.252 °C, and 0.304 °C, respectively. The errors of copper ion concentration are 0.051 ppm, 0.058 ppm, 0.050 ppm, respectively. The performances are adequate for environmental water quality monitoring applications.

## Figures and Tables

**Figure 1 sensors-19-03758-f001:**
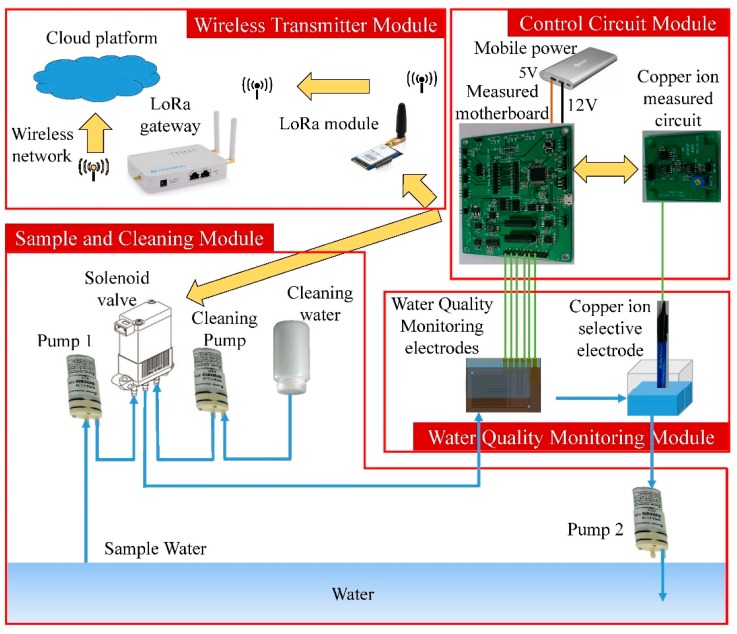
A block diagram of the miniaturized water quality monitoring system.

**Figure 2 sensors-19-03758-f002:**
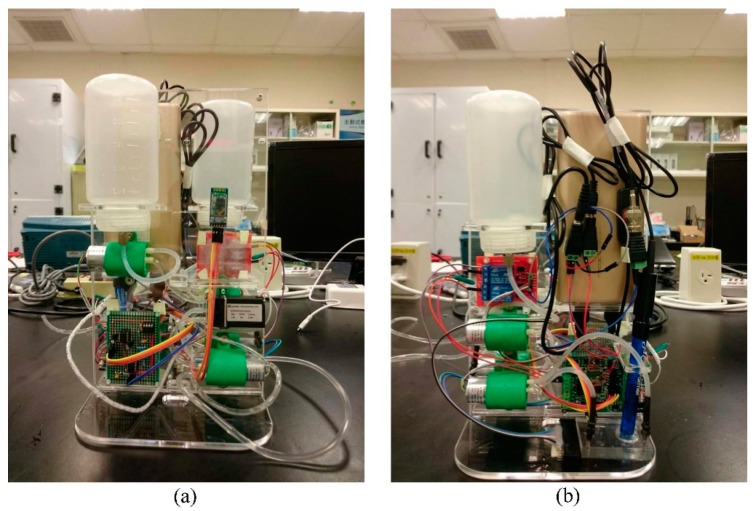
The experimental device diagram of the real water quality monitoring system: (**a**) front view; (**b**) back view.

**Figure 3 sensors-19-03758-f003:**
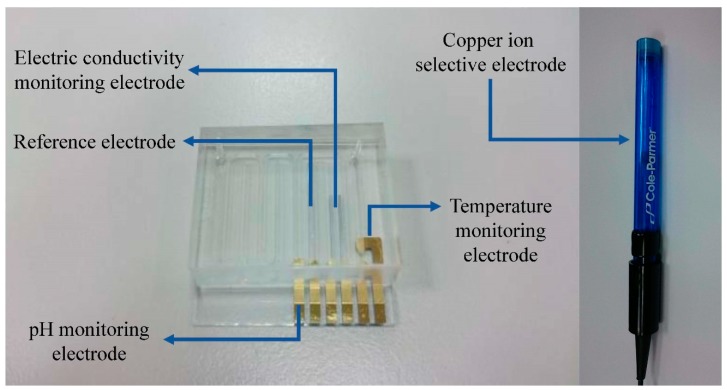
The sensing electrodes of the water quality monitoring module.

**Figure 4 sensors-19-03758-f004:**
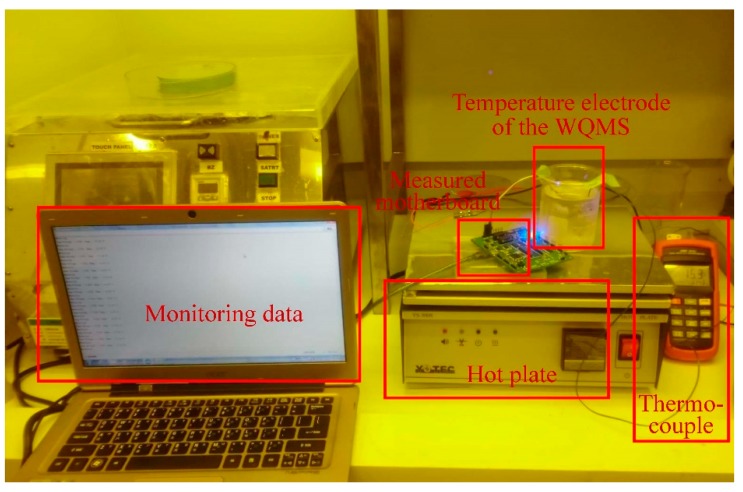
The experimental setup for measuring temperature.

**Figure 5 sensors-19-03758-f005:**
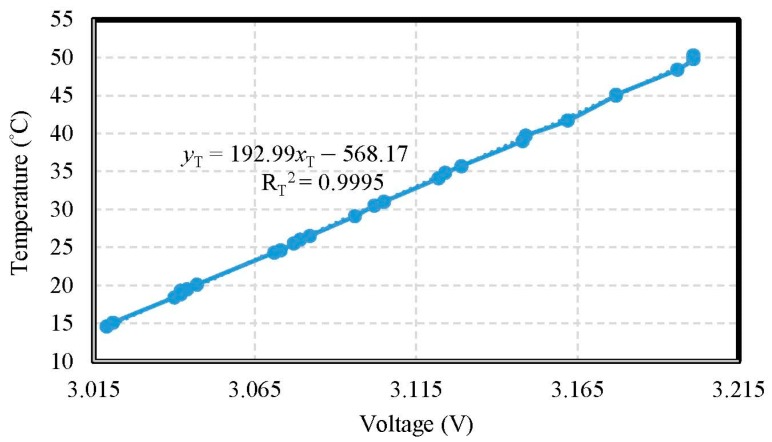
The calibration curve of the temperature.

**Figure 6 sensors-19-03758-f006:**
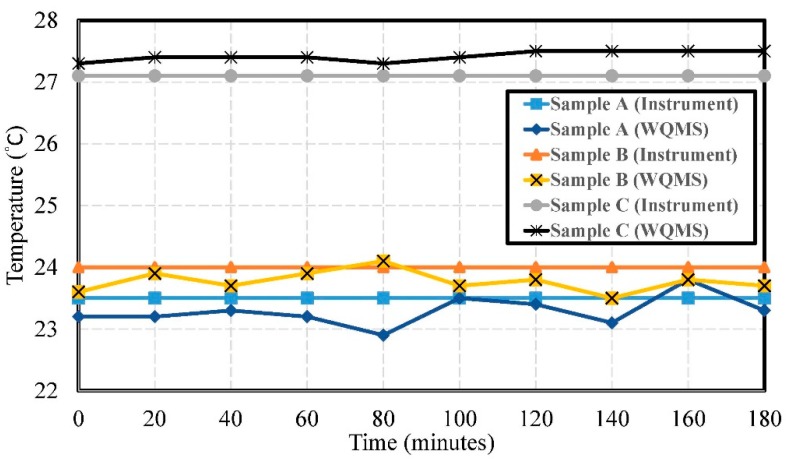
The experimental results of the temperature by using Samples A, B, and C.

**Figure 7 sensors-19-03758-f007:**
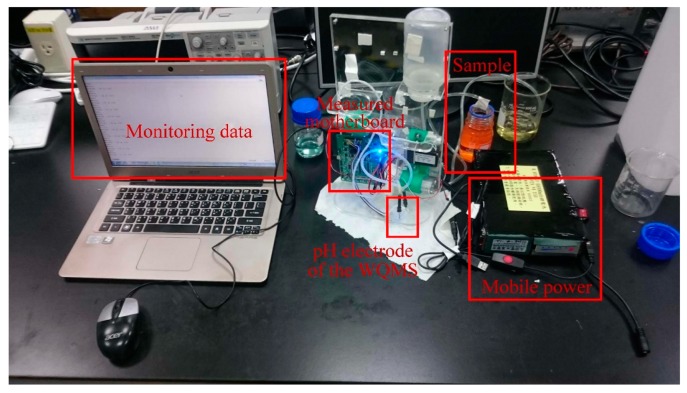
The experimental setup for measuring pH values.

**Figure 8 sensors-19-03758-f008:**
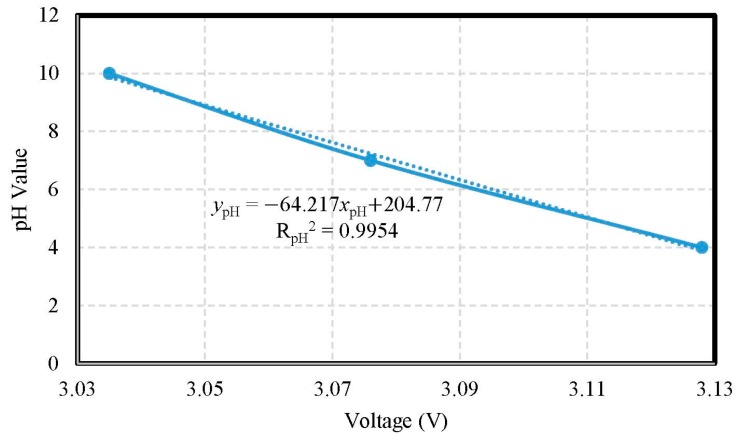
The calibration curve of pH values.

**Figure 9 sensors-19-03758-f009:**
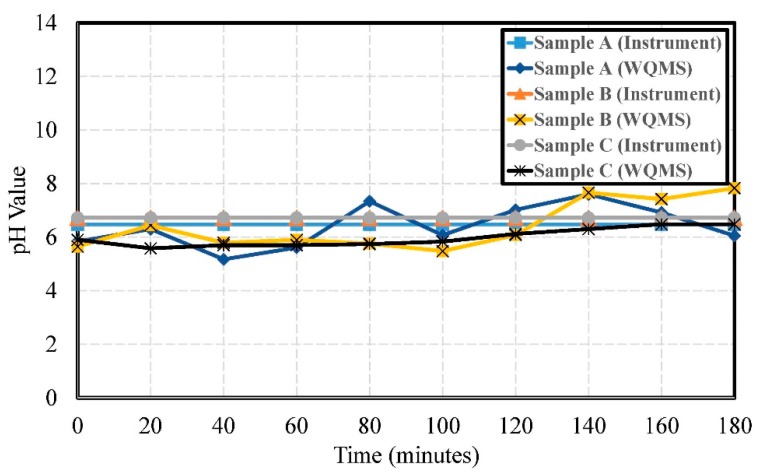
The experimental results of the pH values by using Samples A, B, and C.

**Figure 10 sensors-19-03758-f010:**
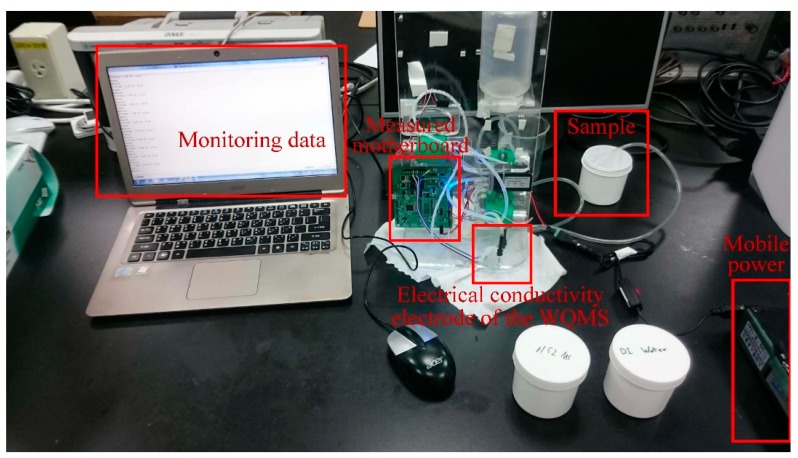
The experimental setup for measuring electrical conductivity.

**Figure 11 sensors-19-03758-f011:**
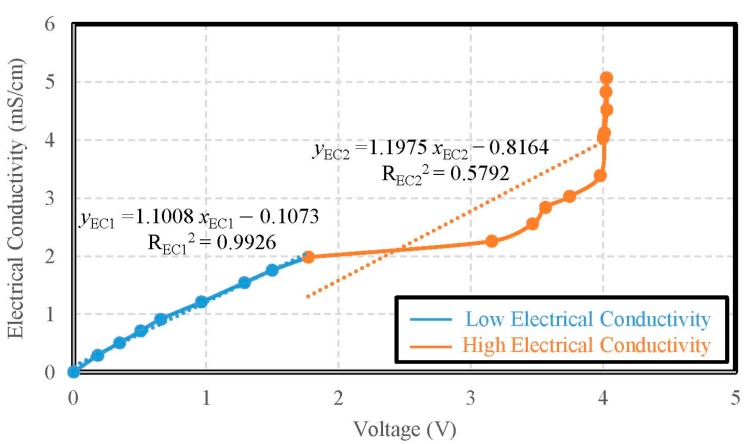
The calibration curve of electric conductivity.

**Figure 12 sensors-19-03758-f012:**
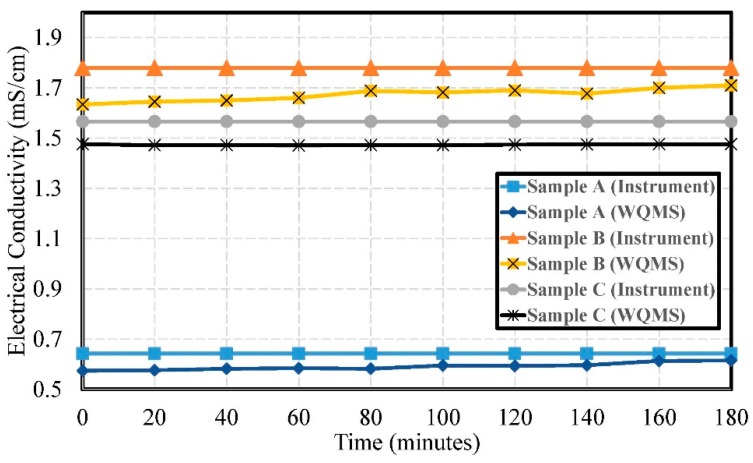
The experimental results of the electrical conductivity by using Samples A, B, and C.

**Figure 13 sensors-19-03758-f013:**
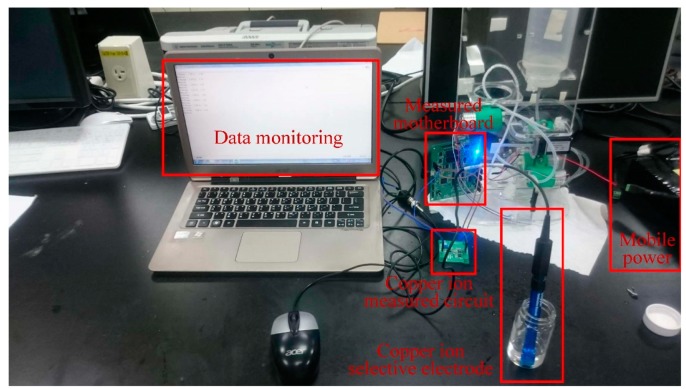
The experimental setup for measuring copper ion concentration.

**Figure 14 sensors-19-03758-f014:**
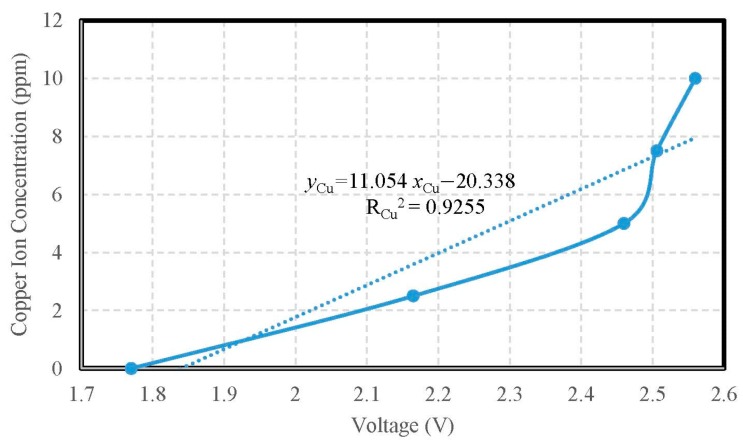
The calibration curve of copper ion concentration.

**Figure 15 sensors-19-03758-f015:**
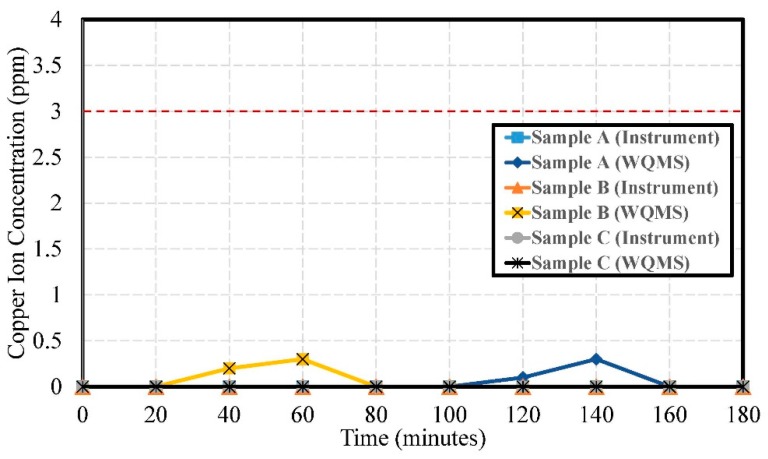
The experimental results of copper ion concentration by using Samples A, B, and C.

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
