# Peer review of "Development of Miniaturized Water Quality Monitoring System Using Wireless Communication"

_sensors, 2019, doi:10.3390/s19173758_

Round 1
Reviewer 1 Report
From the reading of the paper below the following observations are listed:
1.The abstract must be rewritten it is not highlighted which is the advantage or the contributions of the work presented, sensor, instrument?
2. The English style of the paper must be improved.
3. What is the sensitivity of the sensors to measure temperature, pH, electrical conductivity, and cooper ion concentration used in the module, is it possible to determine it?
4. Based on what criteria it is possible to make the following statement "performances are adequate for environmental water quality monitoring applications"
5. In the work the calibration process of the sensors used is not described in detail, please include it
6. The use of Lora technology in the implementation of the IoT system should be clearer, it is only mentioned at the block level in a very general way.
7.We recommended references include the following references:
1. Bo Zhou, Chao Bian, Jianhua Tong and Shanhong Xia, Fabrication of a Miniature Multi-Parameter Sensor Chip for Water Quality Assessment, Sensors 2017, 17, 157; doi:10.3390/s17010157
2. K. Saravanan, E. Anusuya, Raghvendra Kumar, Le Hoang Son, Real-time water quality monitoring using Internet of Things in SCADA, Environmental Monitoring and Assessment September 2018
3. Brandon P.Wong, BrankoKerkez, Real-time environmental sensor data: An
application to water quality using web services, Environmental Modelling & Software Volume 84, October 2016, https://doi.org/10.1016/j.envsoft.2016.07.020.

Reviewer 2 Report
I wish I could give higher marks but sometimes the level of English is impossibly low. It hurts the scientific explanation therefore I was not able to give higher marks in reference to research itself.
However, the research is also missing some crucial parts. First of all Authors should decide whether they want to publish paper about sensor or device. Even if paper will remain as broad as it is, it will require significant improvements in the description of sensors.
Authors have shown time series of measurements at one or two measurements points. We know nothing about key features of sensors such as: linearity, range, and error. Authors did some unknown calculations I believe and provided some values as "errors". But with the method shown in paper these numbers are meaningless.
For example some unknown "instrument" indicated that there was no Copper ions in the water while the presented device provided some non-zero value. It cannot be reasoned from this that the error is below 1 ppm! Perhaps presented device will always show ion concentration at range below 1 ppm even if real value will be much higher. Please excuse me explanation of such basic things in the field of measurements but this is just an example. It applies to all results presented in the paper. Another example: temperature was "about" (sic!) 24 degrees Celsius while presented instrument designed by authors was described as having error below 0.5'C. This is just an artificial number provided on the basis of some averaging that has no mathematical foundation.
If Authors want to discuss errors of these sensors they should do real characteristics in some range and do it many, many times. Averaging of time series at one or two values is not acceptable. Having this it could be only an introduction to discussion of channel cross-talk and influence of other factors on measured value (e.g humidity, temperature).
IoT is added to the title thanks to usage of LoRA. But LoRA itself is described only theoretically in one paragraph and presented in block diagram. As such it provides no added value to presented research. Authors did not even mention what backend was used so I am highly doubtful about the real level of usage of IoT technologies here.
Finally I would love to see more photographs of real device and the system as they were built, and used in this research.
Round 2
Reviewer 2 Report
Dear Authors,
I appreciate improvements as now article looks much more convincing.
Still, please use some help in English before presenting it to publication.
I recommend to reconsider figure 15 and measurements of Cu concentration. It looks like your instrument has not enough sensitivity for the range in which you are trying to operate it. These two peaks shown in the plot may be just some shot noise. Test should be repeated several dozens of times and then analyzed statistically to be reliable.
There is still no IoT in the paper. Using wireless communication like LoRa is not enough to claim IoT label. Please abandon the bandwagon of IoT buzzword and honestly apply title that is more adequate to what is actually presented.
